# Rule-Based Scan-to-BIM Mapping Pipeline in the Plumbing System

**Taewook Kang [1],\*, Shashidhar Patil [2] , Kyubyung Kang [3] , Dan Koo [3] and Jonghoon Kim [4]**

[1] Korea Institute of Civil Engineering and Building Technology, Goyang-si 10223, Korea
[2] Graduate School of Advanced Imaging Science, Multimedia and Film, Chung-Ang University, Seoul 06974, Korea; patil.shashidhar@hotmail.com
[3] Department of Engineering Technology, Indiana University-Purdue University Indianapolis, Indianapolis, IN 46202, USA; kyukang@iu.edu (K.K.); dankoo@iu.edu (D.K.)
[4] Department of Construction Management, University of North Florida, Jacksonville, FL 32224, USA; jongkim@unf.edu
\* Correspondence: laputa99999@gmail.com; Tel.: +82-10-3008-5143

**Abstract:** The number of scan-to-BIM projects that convert scanned data into Building Information Modeling (BIM) for facility management applications in the Mechanical, Electrical and Plumbing (MEP) fields has been increasing. This conversion features an application purpose-oriented process, so the Scan-to-BIM work parameters to be applied vary in each project. Inevitably, a modeler manually adjusts the BIM modeling parameters according to the application purpose, and repeats the Scan-to-BIM process until the desired result is achieved. This repetitive manual process has adverse consequences for project productivity and quality. If the Scan-to-BIM process can be formalized based on predefined rules, the repetitive process in various cases can be automated by re-adjusting only the parameters. In addition, the predefined rule-based Scan-to-BIM pipeline can be stored and reused as a library. This study proposes a rule-based Scan-to-BIM Mapping Pipeline to support application-oriented Scan-to-BIM process automation, variability and reusability. The application target of the proposed pipeline method is the plumbing system that occupies a large number of MEPs. The proposed method was implemented using an automatic generation algorithm, and its effectiveness was verified.

**Keywords:** rule; Scan-to-BIM; mapping; pipeline; plumbing

## 1. Introduction

Recently, for the purpose of facility maintenance, there are increasing cases of introducing reverse engineering technology, which has been mainly studied in the manufacturing field, into the construction field. Since many construction projects are placing orders for BIM, interest in Scan-to-BIM has naturally increased. Scan-to-BIM work has different work parameters depending on the purpose of the project. Inevitably, a BIM modeler manually adjusts the modeling parameters according to the application purpose and performs the Scan-to-BIM process. This process generates repetitive manual work according to the project, leading to work productivity and quality problems.

If the Scan-to-BIM process can be formalized and automated based on predefined rules, this process can be conveniently reused depending on the purpose of the application. This study proposes a method of customizing and recycling the process by formalizing the Scan-to-BIM workflow according to the application purpose, and slightly changing the work parameter. To this end, this study proposes a rule-based Scan-to-BIM Mapping pipeline processing method. The proposed method limits the scope of the plumbing system that occupies the majority in MEP.

This study first analyzes related research trends through extensive literature review and discusses state-of-the-arts and gaps in this field. This study subsequently analyzes the Scan-to-BIM work use-case scenario, the Scan-to-BIM algorithm and related parameters, and conducts a study to formalize them into rules and pipeline structures. After implementing the proposed content with pipelines and algorithms, performance in the various cases is checked.

## 2. Related Works

Research on scanning, Point Cloud Data (PCD) processing, and BIM is on the rise, particularly for application in facility management and construction management.

Among the attempts to digitize MEP information, there has been a study on the creation of connection relationships [1]. This study semi-automatically creates a connection relationship between BIM objects. With the aim of creating a connection relationship between MEPs, an average accuracy of 80% or more was achieved. This method uses a graph data structure to manage connections. However, this study does not use the scan data, but uses the geometric conditions of the already created shapes. A previous study on the creation of 3D objects based on point clouds used deep learning [2]. This study reassembles the 3D shape by determining the building structure type using the PointNet classifier and comparing the similarity with the nearest BIM object in the feature space.

There is a generalized study of the process applied when converting scan data into BIM objects [3]. This study proposes a process and framework through case studies. A study on the implementation of a prototype for indoor service purposes was also reviewed [4]. In this study, a high-level system architecture is proposed as an application study on the point cloud itself.

One study focused on obtaining a 3D model from a 3D point cloud [5]. This study proposes a method to construct a three-dimensional HBIM (Heritage Building Information Modeling) model of a church and stone bridge using the built-in functions of Rhino software. A study on the development of a framework that can be used as a guideline and analyze the Scan-to-BIM process was also reviewed [6]. The present study focuses on the development of guidelines.

In order to utilize BIM in the facility operation stage, a study proposed VIMS (Visualization, Information Modeling, and Simulation) technology using deep learning and computer vision [7]. In this study, a method of linking with BIM objects using data obtained through vision images and deep learning is proposed. A semantic labeling study for Scan-to-BIM focused on creating a mesh using Rhinoceros and Grasshopper and then labeling BIM planar members using the geometric features of the mesh surface [8]. Another study proposed 4D simulation technology to use the scan data for construction management [9]. This study proposes a 4D as-built point cloud simulation method.

One study reviewed the scanning automation planning method in the construction industry [10]. This study defines scan data quality indicators and proposes a scanning process that considers quality management. Another study examined the scanning technology to verify the quality of precast concrete members after completion [11]. This study focuses on the application of scanning technology. A proposed Scan-to-BIM framework using deep learning was also reviewed [12]. This study proposes a deep learning model using PointNet for the segmentation process.

One study examined the quality inspection method for the MEP module constructed in the field [13]. This study proposes a method of extracting some parts of MEP from scan data and comparing it with BIM. Another study probed a quality test method that improved the existing structural safety diagnosis method [14]. This study utilizes scan and BIM technologies for large-scale civil infrastructure inspection. One other study proposed a method of quality inspection using laser scanning to see if the MEP designed with BIM was correctly installed on the site [15]. This research focuses on the development of a method for quality checking laser scans.

A study on a drone-based scan simulation technology confirmed that said technology can detect MEP objects [16]. In this study, simulation was performed in ROS (Robot Operating System) using ORB SLAM (Simultaneous Localization and Mapping) technology.

One study proposed a field data capturing model based on BIM [17]. This study was conducted mainly on literature and technical investigations related to the proposed model. When creating MEP facilities with BIM, a study analyzed requirements and cases for facility management [18]. This study focuses on As-Built MEP modeling work. Another study focused on the effect of reverse engineering based on 3D scan image [19]. This study focuses on the analysis of the effects of Scan-to-BIM reverse engineering applied through real case studies. Another study investigated scan and BIM requirements using point cloud for automated construction progress monitoring [20]. This study identifies building construction quality factors and defines the related point cloud quality variables.

One study proposed a method for Scan-vs-BIM deviation detection [21]. This study is about quality inspection methodology. Another study focused on the Scan-to-BIM methodology [22]. This study points out that Scan-to-BIM is a very difficult and time-consuming task. This study also investigates related automation methods such as tools and technologies. There has been a study to optimize the scan-to-BIM work path planning and scan location [23]. This study focuses on optimizing the scan plan. Another study proposed a 6D (XYZ + RGB)-based Scan-to-BIM methodology [24]. This study improves the accuracy of searching for BIM objects with similar features through 6D data on the interior walls of buildings.

One study proposed a methodology to describe the major steps of the Scan-to-BIM process [25]. This study proposes a classification system, a level of detail (LoD), a scan parameter, a tool, and an analysis method as a framework. There was also a study on the use of scanning technology for HBIM construction [26]. This study focuses on case analysis. Another study focused on converting scan data into graph format [27]. This study focuses on the representation of interconnected geometry. An alternative study on the method of developing a deep learning training model using the point cloud obtained from BIM was also reviewed [28].

There was a study on the management of MEP facilities based on 3D reverse engineering [29]. This study focuses on facility management technology using scan and MEP BIM. Another study delved into the automatic inspection method between BIM and scan data [30]. This study focuses on the automatic inspection technology between the model and the point cloud.

The previously investigated studies deal largely with Scan-to-BIM guidelines for specific applications, quality control methods, work performance methodology including tools, and generation algorithms for some shape types. This study proposes a rule-based Scan-to-BIM Mapping Pipeline method that can support application-oriented Scan-to-BIM process variability, automation, and recycling.

## 3. Scan-to-BIM Mapping Scenario Analysis

### 3.1. Pipeline Concept

The concept of a pipeline refers to a structure or a set of data processing elements in which the output of the prior processing step is connected to the input of the next. These connected pipeline steps become a standardized process that can be easily replicated for other projects with only minor modifications in regular and formal parameters.

Since multiple predefined pipelines can be executed collaboratively, performance can be much improved by processing a large amount of data in parallel. If a pipeline well determines a standardized input/output format, parameters, and logistics for execution steps, reusability rates can be improved by modifying the parameters of the predefined pipeline according to the purpose of use. This can be a meaningful way to increase productivity in Scan-to-BIM projects with many repetitive tasks. Figure 1 shows a conceptual diagram of the Scan-to-BIM Mapping Pipeline proposed in this study.

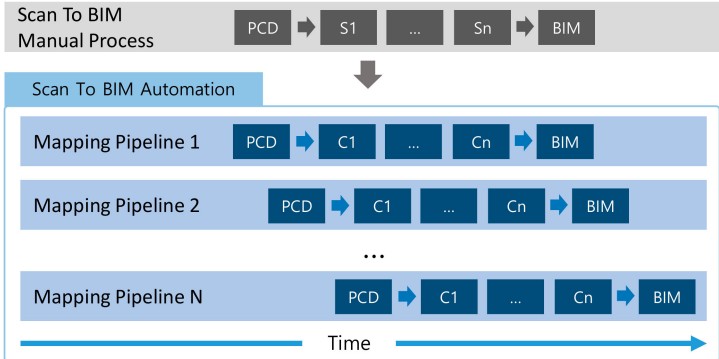

**Figure 1.** Scan-to-BIM Mapping Pipeline Concept Diagram.

It is important to analyze a Scan-to-BIM use-case scenario and define its processing steps in designing a pipeline structure. Through this, elements constituting a process can be mapped to the pipeline components. Each step in the process transforms the data through a variety of tools, techniques, and manual interventions, and sends it to the next step. At this stage, a modeler determines steps for automation and analyzes what data inputs/outputs and operation parameters are required for the pipeline structure. Data input/output types and operation parameters can be defined as generalizable rules. Scan-to-BIM rules can be stored in a library through predefined pipelines and parameters, allowing them to be reused in similar use-cases.

A scenario analysis of Scan-to-BIM use-cases, process component definition, data input/output format, and parameter normalization studies is discussed in the following subsections.

*3.2. Use-Case Scenario Analysis*

A Scan-to-BIM project for MEP with Plumbing System was performed to validate the proposed rule-based Scan-to-BIM Mapping Pipeline method. The target of the scan is the MEP facility of the K-Institute's Zero Energy Building. The number of scanned point clouds is 41,790,655, and the capacity is 1,504,463,580 bytes. For reference, the point cloud is composed of coordinate values, intensity, RGB values, etc. as follows.

$$PCD_{volume} = PCD_{count} \cdot (point_{size} + RGB_{size}) \tag{1}$$

The main elements that compose the Scan-to-BIM process are as follows.

- S1. Field trip and planning
- S2. Target installation for matching
- S3. Field scanning
- S4. Data matching
- S5. Filtering
- S6. Segmentation
- S7. Modelling
- S8. Scan-to-BIM Quality Check Report

Among these elements, the steps that can be relatively automated are S4, S5, S6, and S7. In this study, S5, S6, S7 are targeted for automation, while S4 is not. Figure 2 shows the use-case diagram. Each use-case can be defined as a Scan-to-BIM pipeline component.

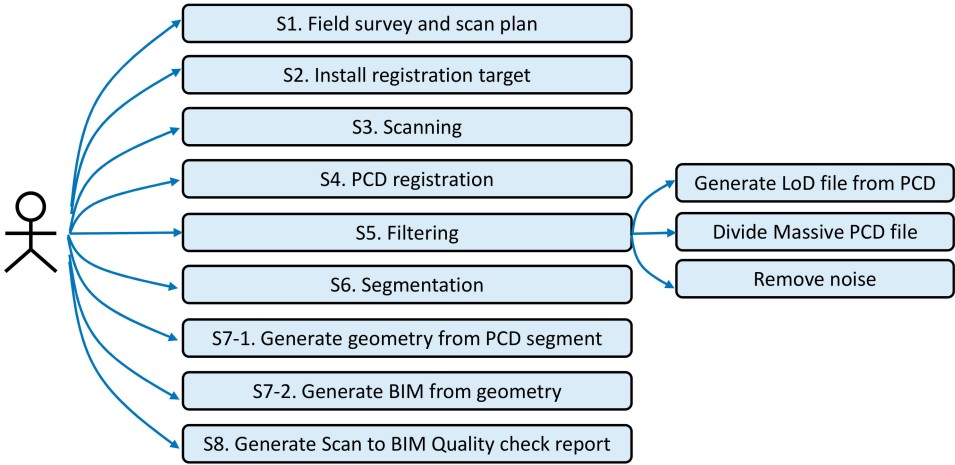

**Figure 2.** Use-case diagram.

*3.3. Pipeline Component Design and Input/Output Data Normalization*

In order to design the structure of the Scan-to-BIM pipeline, it is necessary to analyze the input/output data items and types related to each step. This study formalized component and scan data input/output types based on the use-case scenario analyzed in Section 4.2. Figure 3 shows the relationship and input/output data items for each step of the Scan-to-BIM pipeline design.

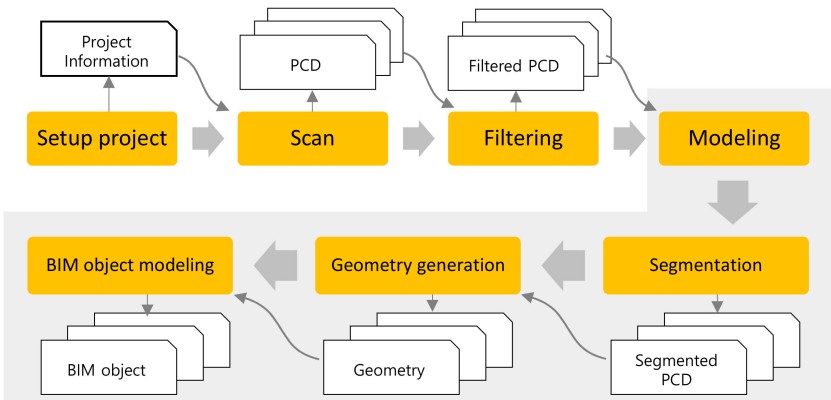

**Figure 3.** Scan-to-BIM Pipeline Component and Input/output Data.

When the Scan-to-BIM use-case scenario is generalized, each step of the process can be divided into Scan-to-BIM project setting, scanning, filtering, modeling, segmentation, shape creation, and BIM object modeling steps. In every step, the scanned PCD is received and the PCD is processed until a BIM object is obtained. Each step has a specific type of data item and an input/output relationship. For example, in the point cloud filtering step, a PCD file is input based on a working parameter related to point cloud noise removal, and the filtered PCD file is generated. The geometry generation step yields geometry by receiving segmented PCD. Table 1 summarizes the pipeline components and input/output data derivation.

Pipeline data management becomes convenient if the connection relationship between pipeline components and input/output data items can be structured. By grouping input/output data items related to components, independent reuse is possible for each component. Taking this into account, the relationship between component and data item is structured in a hierarchy as shown in Figure 4.

**Table 1.** Scan-to-BIM pipeline component and input/output data derivation (* = multiple).

| Scan-to-BIM Pipeline Component | Data Item | Data Format |
|---|---|---|
| C1. Setup project | D1. Project information file | {name, description} |
| C2. Scan | D2. PCD file | {x, y, z, I, RGB} * |
| C3. Grid generation | C3. PCD file | {grid.ID, x, y, z, I, RGB} * |
| C4. Level of Detail (LoD) | D4. PCD file | PCD files |
| C5. Filtering | D5. Filtered PCD file | PCD files |
| C6. Segmentation | D6. Segmented PCD | {segment.ID, x, y, z, I, RGB} |
| C7. Geometry generation | D7. Geometry data | {geometry.ID, type, dimension *} Dimension = {name, value} |
| C8. BIM object generation | D8. BIM object data | {BIM.object.ID, type, dimension *, property*} Property = {name, value} |

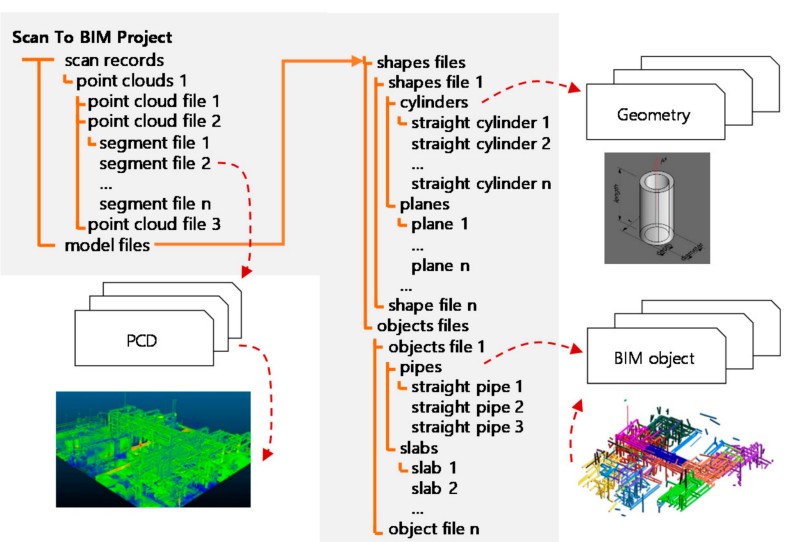

**Figure 4.** Scan-to-BIM Pipeline Component Dataset Hierarchy Structure Definition.

## 4. Pipeline Component Parameter and Rule Design

### 4.1. Scan-to-BIM Component Algorithm and Parameters

To design Scan-to-BIM component operation parameters and rules, this section investigates the algorithms and parameters of each component in Table 1. If the working parameters can be extracted and ordered, the predefined Scan-to-BIM pipeline can be replicated with only a few parameter modifications. The parameter value must be able to be determined differently according to the purpose of Scan-to-BIM. In other words, Scan-to-BIM configuration with parameter setting is purpose-oriented. As shown in Figure 5, the modeler sets parameters to suit the purpose or reuses them in the parameters library. This is because the dimensions and properties of the modeled BIM object are different depending on the use case.

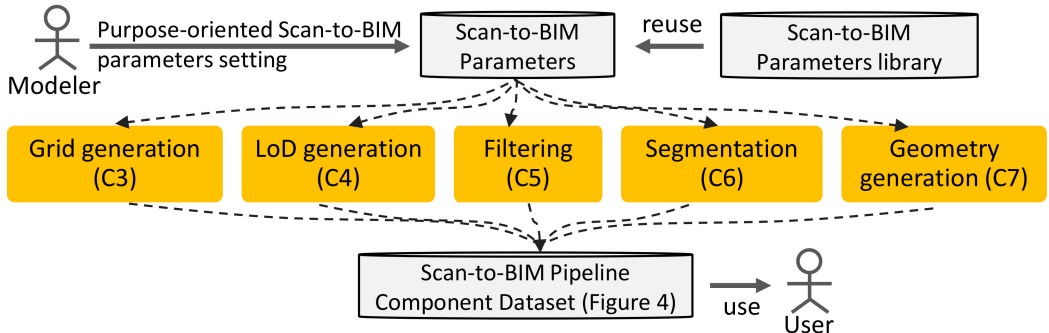

**Figure 5.** Purpose-oriented Scan-to-BIM Parameters Setting related to Pipeline Component.

### 4.1.1. Grid Generation(C3), LoD(C4) Parameters

If a large amount of point cloud data is used for a Scan-to-BIM project, a processing problem on a computer arises. In general, the data size obtained by matching point groups obtained from LiDAR is tens of millions to more than billions of points. Not all projects need this high-precision point cloud. In general, point clouds are used by converting precision according to the purpose. Even after converting the LoD of the point cloud to a low level, if the computer is not capable to process the data, the point cloud should be divided into regions. Grids are used for segmentation. The grid size can be defined by specifying the x, y, and z axis spacing in the 3D coordinate system.

$$parameter_{grid} = \left\{size_x, \; size_y, \; size_z\right\} \tag{2}$$

LoD processing transforms the input point group with a density equal to or lower than the specified number of points. LoD can be expressed as the number of points in a specific area like resolution. LoD processing generally uses a quadtree, octree spatial indexing technique to divide a space and calculates the average point and center point of the point group of the divided area. The related parameters are as follows.

$$parameter_{LoD} = \{size, \; point_{count}\} \tag{3}$$

### 4.1.2. Filtering(C5) Parameters

When unwanted areas or point clouds are acquired during the scan step, the process of removing these noise data directly affects the quality of the subsequent work. Automatic filtering is a method to remove the region after determining the region to be removed using a clustering algorithm such as k-NN (k-nearest neighbor). When a small number of point groups are separated by more than a certain distance from the dense point group, or when point group data outside the region of interest are acquired, these point groups will be removed. Equations (4)–(6) express parameters for these point groups.

$$parameter_{filter} = \left\{filter_{k-NN}, \; filter_{bounding\_box}\right\} \tag{4}$$

$$filter_{k-NN} = \{distance, \; point_{min\_count}\} \tag{5}$$

$$filter_{bounding\_box} = \{inner \big| outer, \; x1, \; y1, \; z1, \; x2, \; y2, \; z2\} \tag{6}$$

### 4.1.3. Segmentation(C6), Geometry Generation(C7) Parameters

Segmentation is processed in two stages. The first step is to perform segmentation based on the point where the curvature is sharply divided, and the second step is to calculate a point group with parameters that fit the pipe shape.

The segmentation uses a region growing algorithm based on curvature similarity. This method employs the local surface normal and connectivity of the point group as shown in the following equation.

$$\|n_0 \cdot n_1\| > \cos(\theta_{th}) \tag{7}$$

where, $n_0$ is the normal of each point, $n_1$ is the angle between the current point and the normal of the surrounding points, and $\theta_{th}$ is an allowable angle. The allowable angle can be expressed as a parameter as follows.

$$parameter_{segmentation} = \{\theta_{th}\} \tag{8}$$

The second-stage segmentation uses RANSAC (Random Sample Consensus) algorithm. Pipe shape can be expressed by cylinder geometry. A cylinder can be defined by its axis, diameter, radius, and length. The mathematical simplification of the cylinder model is as follows.

$$(x - a)^2 + (y - b)^2 = r^2 \tag{9}$$

This is expressed as a parameter as follows.

$$parameter_{cylinder\_segmentation} = \{radius, \ legnth\} \tag{10}$$

The pipe shape extraction algorithm model is as follows. Each step can be repeated until the points satisfy the cylinder shape.

- C1. $P_{1,\ 2,3} = \{point \mid point_{random} \text{in PCD}\}$
- C2. $parameter_{plane} = \{plane_{a,\ b,c} \mid PCA(P_{1,\ 2,3})\}$
- C3. $P_c = \{point \mid \text{center point of circle from } P_{1,\ 2,3},\ plane_{a,\ b,c}\}$
- C4. $r = radius(P_c,\ P_{1,\ 2,3}\}$
- C5. $axis_{vector} = axis(plane_{a,\ b,c}, P_c\}$
- C6. $geometry_{cylinder} = cylinder(axis_{vector}, P_c,\ r\}$
- C7. $number_{inlier} = \{count \mid PCD \ in \ geometry_{cylinder} \text{with tolerance}\}$

Here, $P_{1,\ 2,3}$ represents a seed point for cylinder search. C1 selects and returns a random point from the PCD. C2 uses principal component analysis (PCA) to obtain plane equation coefficients from a given PCD. Using the plane and seed points, a circle can be constructed, and the center point $P_c$ of the circle and the radius r can be obtained together. By acquiring the axis using the plane and $P_c$, the cylinder shape can be calculated. All the included points are searched by applying tolerance to the calculated cylinder shape. The number of searched points is returned as $number_{inlier}$. Repeat steps up to C1–7 so that $\max(number_{inlier})$.

The Elbow pipe extracts the straight pipe extracted from the point cloud data divided by segmentation, compares the points of each end point between the straight pipes, and extracts the point of the closest pipe as the connected pipe. Elbow pipe connection information is extracted through the direction information of the extracted pipes and the calculation of the intersection point.

*4.2. BIM Object Generation and Mapping Rule*

If the geometry is extracted from PCD, the shape must be mapped to a BIM object that meets the user's requirements. In general, since the shape extracted from PCD contains errors, the dimensions cannot be used as they are. In addition, since the geometry does not contain attribute information such as material, it is difficult to use it. Therefore, it is necessary to map the shape to the object based on the intended use. To solve this problem, this study proposes a method of regularizing BIM object mapping. By pre-defining the object type, material properties, dimensions, etc. required when the

geometry is objectified, the BIM modeling task can be automated in a way that is more effective than when manual human intervention is used.

The BIM object mapping rule is defined as follows using the previously defined task parameter and decision tree. If the rule including the condition is satisfied, the geometry can be converted into a BIM object using the specified $BIM_{object.type}$, $BIM_{object.properties}$.

Scan-to-BIM

$$
\begin{aligned}
&\text{mapping rule } = \left\{\text{condition}^*, BIM_{object.type}, \ BIM_{object.properties}\right\} \\
&\text{condition } = \{\text{operator, logic}\}, \text{ operator } = \{\text{parameter, operator, value}\} \\
&\text{parameter } = \{\text{Scan} - \text{to} - \text{BIM component's algorithm parameter}\} \\
&\text{compare } = \{'<', \ '>', \ '<=', \ '>=', \ '='\}, \text{ logic } = \{'AND', \ 'OR', \ ''\} \\
&BIM_{object.type} = \text{ BIM object type}, \ BIM_{object.properties} = \text{ BIM object property set}
\end{aligned}
\tag{11}
$$

## 5. Scan-to-BIM Mapping Pipeline Component Design

### 5.1. Pipeline Structure Requirement

The entire pipeline structure is designed based on the previously analyzed Scan-to-BIM scenario, use-case, process, input/output data structure, parameter definition, and mapping rules. Scan-to-BIM mapping pipeline should satisfy the following requirements considering scalability, reusability, and parallel processing.

- R1. Scan-to-BIM pipeline components have independent input/output structures.
- R2. The input/output formats of the Scan-to-BIM pipeline components are compatible with each other.
- R3. It supports a combination of Scan-to-BIM pipeline workflow.
- R4. It supports parallel processing of the Scan-to-BIM pipeline workflow.
- R5. It supports large-scale Scan Data Processing.
- R6. Supports parameter variability when executing Scan-to-BIM pipeline components.

Among them, the requirements R1, R2, R5, and R6 can be satisfied using the previously defined input/output data structure, parameter definition and mapping rules.

### 5.2. Pipeline Structure Design

Pipeline design uses the concept of software design patterns. The design pattern is a pattern used to solve problems that often occur in object-oriented design. Each component in the pipeline has an input/output interface, and the interface uses the simplest file system.

Container Pattern and Iterator Pattern are used to define the workflow of the pipeline. Command Pattern is used to manage and execute the defined workflow. Each component makes it possible to call a script so that parameters can be executed variably. Considering the requirements R3 and R4, design the Scan-to-BIM mapping pipeline structure using UML (Unified Modeling Language) as shown in Figure 6 and Table 2.

Components call external scripts, converting inputs into outputs. Scripts can define steps to handle input and output from the user's point of view.

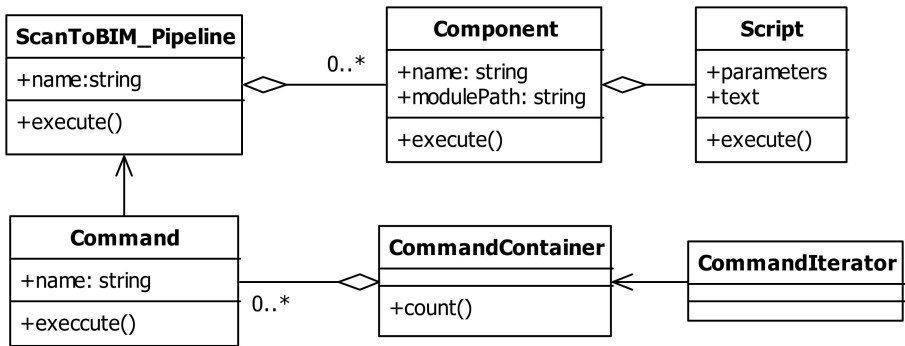

**Figure 6.** Scan-to-BIM Pipeline Architecture (UML).

**Table 2.** Scan-to-BIM class role definition.

| Class | Role |
|---|---|
| ScanToBIM_Pipeline | Manages the components that make up the pipeline.<br>name:string = pipeline name<br>execute() = pipeline execution function |
| Component | Define components for each stage of the pipeline workflow.<br>name:string = pipline component name<br>modulePath: string = component module path<br>execute() = component execution function |
| Script | Manages a script that defines how each stage of the pipeline is executed.<br>parameters = component parameters for execution<br>execute() = script execution function |
| CommandContainer | Manages the recycling of commands that run pipeline workflows. |
| Command | Manages commands to run pipeline workflows.<br>name:string = command name<br>execute() = command execution function |
| CommandIterator | List of commands. |

### 5.3. Pipeline Component Definition

Redefine the previously analyzed Scan-to-BIM Process Step as a component. The component is defined by a deriving Component class of Scan-to-BIM Pipeline Architecture. Through this, it is possible to increase the extensibility of the pipeline component. Figure 7 shows the overall architecture of the Scan-to-BIM pipeline, while the role of the class components and definitions are summarized in Table 3.

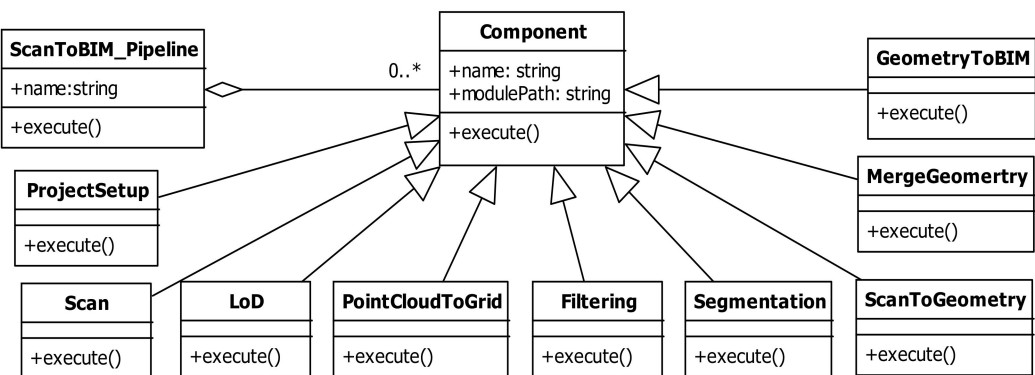

**Figure 7.** Scan-to-BIM Pipeline Architecture (UML).

**Table 3.** Scan-to-BIM pipeline component definition (* = multiple).

| Component | Input | Output | Responsibility |
|---|---|---|---|
| ProjectSetup | Name description position | project setup file | Scan-to-BIM project setting parameter definition. |
| LoD | PCD file resolution (mm) | PCD file | Point cloud resolution is adjusted to a level appropriate for the purpose of use for processing large amounts of scan data. |
| PointCloudToGrid | PCD grid = {x, y, z} | PCD files | Split point cloud data into grids for processing large-scale scan data. |
| Filtering | PCD Noise = {density, kNN distance} clipbox = {x1, y1, z1, x2, y2, z2} | PCD | Removes noise from point groups within the divided grid. The removal method depends on the purpose of use and the scanning environment. The definition of the noise removal parameter specifies the distance between the low-density point group and the other point groups by kNN algorithm. The target point cloud area to be used is designated as a boundary box. |
| Segmentation | PCD cylinder = {min radius, max radius, min length, min curvature, max curvature} | Segmented PCD files parameter | To extract the cylinder shape from the point group, this study defines the features for the point group. Specify the minimum radius, maximum radius and length of the cylinder, which are the characteristics of the pipe shape. Also, the curvature of the points constituting the cylinder is specified. |
| ScanToGeometry | Segmented PCD Segmentation.parameter pipe = {radius, tolerance} * | geometry files | Convert pipe segment point cloud to geometry. Pipe shape dimensions are adjusted to a value corresponding to a radius within a given tolerance considering noise. |
| MergeGeometry | geometry files distance tolerance | geometry file | Segmented pipe shapes are merged with a given tolerance and merged into geometry. |
| GeometryToBIM | geometry fileobject = {type, property*} property = {name, value} | BIM file | Converts geometry files to BIM object files. Defines type and property information to convert to BIM object file. Property information consists of a name and a value. |

Segmentation creates a segmented point cloud that is required to define a cylinder shape parameter. The previously defined Scan-to-BIM pipeline component enables the definition of workflow from the user's point of view.

### 5.4. Scan-to-BIM Data Structure

Scan-to-BIM is a process of converting a given PCD according to the purpose of use and finally mapping it to a shape and a BIM object. Therefore, PCD becomes the core data structure in the pipeline components. The point data structure constituting the PCD may include RGB and intensity values in addition to x, y, and z coordinate values. Points are added as values and attributes calculated for segmentation, etc. and are passed through the pipeline process. For example, in order to calculate the curvature of a point, a normal vector must be calculated. Figure 8 shows a proposed data structure. The roles of major classes are summarized in Table 4.

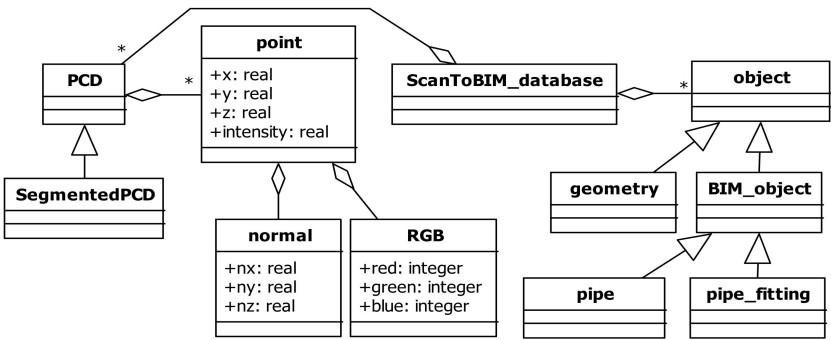

**Figure 8.** Scan-to-BIM Data Structure.

**Table 4.** Scan-to-BIM data structure class definition.

| Class | Role |
|---|---|
| point | It manages x, y, z real coordinate values and reflection intensity. |
| normal | It manages the normal vectors nx, ny, and nz required for point curvature and segmentation. |
| RGB | If the point has color values, this data is managed. Each color channel value ranges from 0 to 255. |
| BIM_object | Manages BIM object information mapped in the shape. |

## 6. Case Study and Performance Analysis

This study conducted a case study to verify the performance of the proposed Scan-to-BIM Mapping Pipeline. The pipeline components and algorithms were developed using C++. The pipeline component can be executed from the console, and a plug-in was developed in Autodesk Revit to invoke and execute the Scan-to-BIM mapping pipeline. A separate viewer was developed using OpenGL to check the PCD segment result. The data used in the test were acquired using terrestrial LiDAR. Scan data is acquired from plant and building MEP. Large-capacity data processing, incomplete PCD processing test, pipe object creation, and comparison tests with programs mainly used in the industry were performed.

### 6.1. Massive PCD Processing Pipeline Performance Test

To test the massive data processing performance of the pipeline, the PCD Preprocessing (PCD-P) time and PCD Visualization (PCD-V) performance were checked. The test data yielded 41,790,655 points via laser scanning the MEP facility of the Korea Institute of Construction Technology. The total point cloud capacity is 1,504,463,580 bytes. Four datasets (A = 100,000, B = 1 million, C = 3 million, D = original) were prepared by adding the LoD stage to the pipeline. For reference, the computer specifications tested are Intel Core i5, quad-core and 4G memory without GPU.

To test the data preprocessing performance of the pipeline, a grid transformation step and a segmentation step were added to the pipeline. Segmentation includes point-by-point normal vector calculation, RANSAC processing time, LoD processing, and grid space division processing. PCD visualization performance includes point cloud search and rendering time according to camera view.

PCD-P pipeline = {LoD, Grid data processing, Normal vector calculation, RANSAC}

PCD-V pipeline = {Point cloud searching, Rendering}

The PCD-P standard deviation among the four datasets is 435.8, whereas the PCD-V standard deviation is only 0.004 (shown in Table 5). For this reason, this study deduces that it is effective to store pipeline processing data with a large amount of calculation in advance, such as normal vector calculation, and load it into memory when necessary.

**Table 5.** PCD processing and rendering performance.

| Case | 1 | 2 | 3 | 4 | Average | STD |
|---|---|---|---|---|---|---|
| Data Preprocessing (sec) | 3.58 | 21.93 | 146.21 | 919.5 | 272.8 | 435.8 |
| Data Visualization (sec) | 0.008 | 0.008 | 0.015 | 0.016 | 0.012 | 0.004 |

### 6.2. Pipe Object Mapping Pipeline Performance Test in Imperfect PCD

Most of the point cloud data collected through a laser scanner and photographic images in the actual field result in incomplete point data. In order to test Scan to BIM mapping performance in an incomplete PCD, Scan-to-BIM mapping pipeline parameters and rules were defined. The accuracy of pipe creation is verified by overlapping and comparing the mapped pipe object and point cloud data. Furthermore, this study compared the results with commercial reverse engineering software to further check the mapping performance.

### 6.2.1. Outdoor Plant Data Test Acquired by Drone Photogrammetry

Point cloud data was acquired after filming the outdoor plant facility of the Korea Institute of Construction Technology using a drone. For photogrammetry, 74 photos were taken using a DJI drone, and point clouds were created using Pix4D. The image resolution is 4000 × 3000 pixels. The drone picture is an incomplete PCD with only the side and top, and no lower data. The total number of points is 2,927,688, and 586,466 after LoD conversion. Mapping rules and parameters for mapping this data were defined and applied to the pipe objects. The number of segments extracted was 30 (takes 1.94 min). A total of 21 linear cylinders were mapped (21.55 s) (Figure 9). Other commercial software such as L company product and E company product software had an error in shape extraction, so the result could not be confirmed. In the case of incomplete data, it was difficult to obtain proper results when the function was executed with default settings.

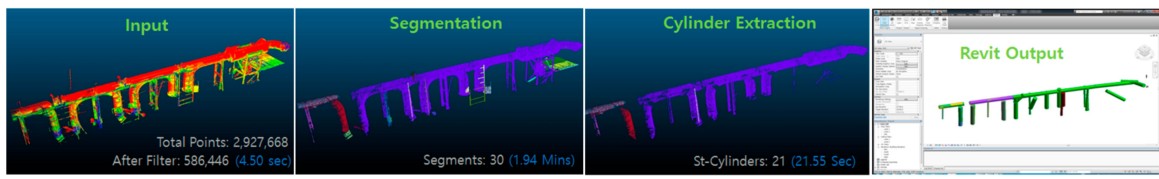

**Figure 9.** PCD BIM Object Extraction results and error.

### 6.2.2. Testing of Indoor MEP Facility Data Acquired with LiDAR

To test the ability to extract numerous pipe objects, complex MEP facilities in the building were scanned with LiDAR. There were three scanned MEP locations, divided into A, B, and C. The number of points are 480,220 (9.15 MB), 364,340 (6.94 MB), and 807,509 (15.4 MB), respectively (Figure 10).

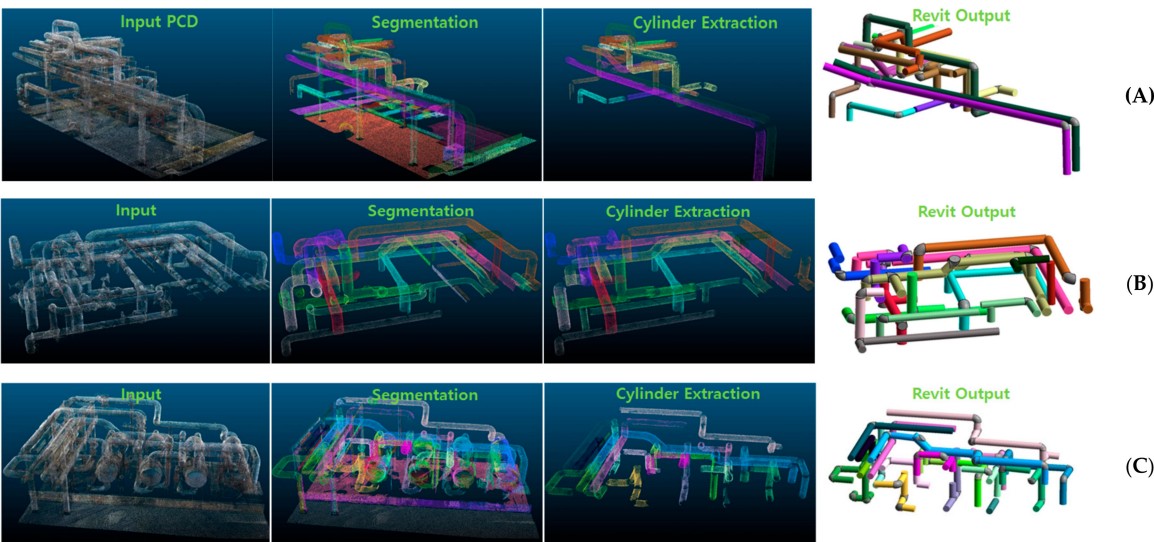

**Figure 10.** Scan-to-BIM Mapping Pipeline Results (**A**,**B**,**C** cases).

Table 6 shows the performance of the Scan-to-BIM mapping pipeline. The performance comparison of linear and Elbow pipe mapping was done with the results of manual work. Each performance averaged 87.8% and 75.5%, respectively.

**Table 6.** Scan-to-BIM mapping pipeline performance (count, %).

| Case | PCD | Filtering, LOD(MB) | Segmentation | Straight Pipe | | Elbow-Type Pipe | |
|---|---|---|---|---|---|---|---|
| | | | | Manual | Automatic | Manual | Automatic |
| A | 480,220 | 331,595(5.05) | 97 | 60 | 51(85.0%) | 40 | 29(72.5%) |
| B | 364,340 | 122,166(1.86) | 35 | 60 | 55(91.7%) | 41 | 27(65.9%) |
| C | 807,509 | 343,056(5.23) | 175 | 90 | 78(86.7%) | 59 | 52(88.1%) |
| Average | 550,690 | 265,606(4.05) | 102.3 | 70.0 | 61.3(87.8%) | 46.7 | 36.0(75.5%) |
| STD | 229,835 | 124,355(1.90) | 70.2 | 17.3 | 14.6(3.48%) | 10.7 | 13.9(11.4%) |

6.2.3. Pipe Object Mapping Pipeline Performance Test in Massive PCD

To test the pipe object mapping performance in massive PCD, the factory plant equipment was scanned with LiDAR. The scanned PCD size is 89,449,621 (2GB) (Figure 11). After calculating the total size of the point group, the grid sizes divided by three on the X-Y-Z axis were used to generate voxels. Voxels were used to improve PCD calculation performance. Down-sampling was performed for LoD processing, and the parameter value was 2 cm. For reference, the pipe radius required was between 20 and 50 cm. The difference in diameter between these pipes was 6 cm. When mapping the extracted geometries to BIM objects, they were converted into BIM objects using Scan-to-BIM mapping ruleset. In addition, the parameters such as grid size or LoD were chosen considering these situations.

The final data is approximately 85% of pipe objects extracted from the point cloud data.

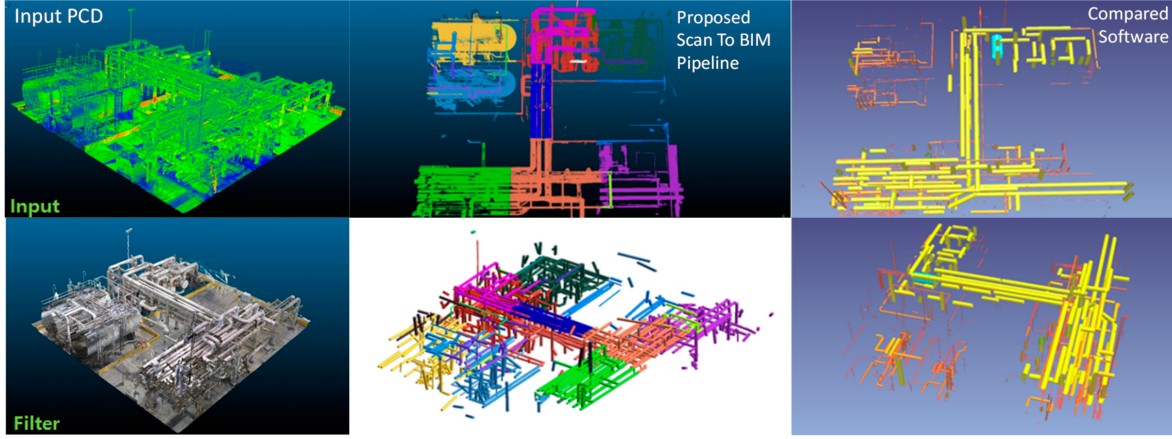

**Figure 11.** Massive PCD Processing Results (Input PCD, Proposed Scan-to-BIM Pipeline, Compared Software).

This paper checked the object mapping performance between the proposed pipeline method and Company E's software. Company E products are mainly used in the industry for automatic pipe extraction. As shown in Table 7, the pipe object extraction performance of the proposed method is improved by 104.4%, and processing speed is improved by 687.5% compared to the software.

**Table 7.** Pipe object mapping performance results.

| Program | PCD (GB) | Filtering (MB) | Pipe (A/B%) | Speed (min)(B/A%) |
|---|---|---|---|---|
| A. Proposed | 89,449,621 (2GB) | 8,994,608 (137MB) | 1099 | 8 |
| B. Compared | 89,449,621 (2GB) | - | 1053 | 55 |
| Difference (A−B) | - | - | 46 (104.4%) | 47 (687.5%) |

Pipe extraction performance errors mainly occurred in small and complex pipes. The proposed method can precisely set the parameters of the Scan-to-BIM mapping pipeline, so that it is possible to

obtain precision that is difficult to set in the comparison software. For reference, this study did not compare the overall functionality of the comparison software that is not related to the scope of this study. Based on the previously obtained data, the time required for work was compared as shown in Table 8. It is assumed that the scan-to-BIM mapping work of the same data size was performed with eight projects. In this case, 376 min per day and 7520 min per month are saved when measured against the comparison software.

**Table 8.** Pipe object mapping performance comparison with manual process.

| Project Size (GB) | Project Count | Performance | | Saved | |
|---|---|---|---|---|---|
| | | Proposed | Compared | 1 Day (8 h) | 1 Month (20 Days) |
| 2 | 8 | 64 (1 h 4 m) | 440 (7 h 20 m) | −376 (6 h 16 m) | −7520 (125 h 20 m ≈ 15 days) |

#### 6.2.4. Non-linear Elbow Pipe Mapping Performance Test

This study tested the right-angle pipe mapping performance, which corresponds to the tolerance range. As shown in Figure 12, most of the pipes were mapped over 90%. Elbow pipe start and end connection points have improved precision by adjusting the parameters of the pipeline component to suit the purpose.

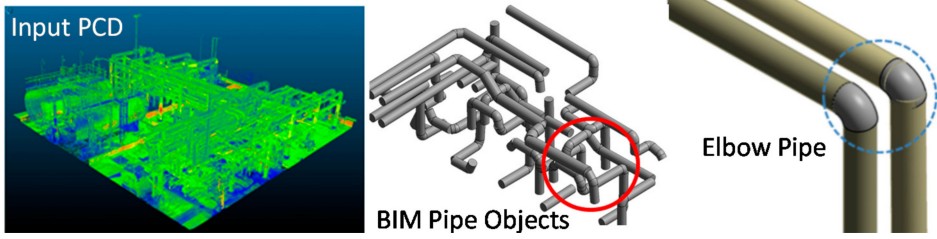

**Figure 12.** Elbow-Type Pipe Object Mapping Results.

### 6.3. BIM Object Mapping Test

The shape modeled in the Scan-to-BIM project must be mapped to an object including the attributes. For this, this study tested it using the previously defined BIM object mapping rule. By using the rule and the pipe shape parameter created in the pipeline, a BIM object that matches the rule is created in the module implemented with the Autodesk Revit plug-in. Figure 13 shows the result of BIM object property mapping.

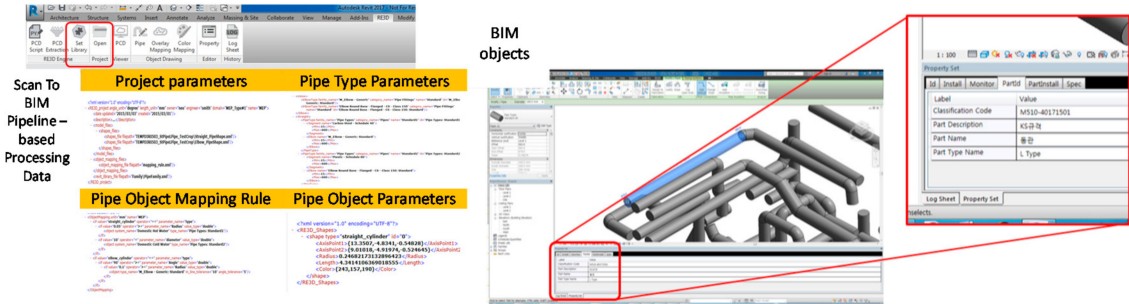

**Figure 13.** BIM Object Property Mapping Results in Revit.

## 7. Conclusions

This paper investigated current research trends to propose the structure of the proposed Scan-to-BIM mapping pipeline. Subsequently, this study analyzed a use-case scenario for the Scan-to-BIM work, the Scan-to-BIM algorithm and related parameters, and formulated them into rules and pipeline structures. The proposed method was implemented with a Scan-to-BIM mapping pipeline and algorithm. Large-capacity PCD, incomplete PCD, outdoor plant, and indoor building MEP data were acquired using drone photogrammetry and LiDAR scan to confirm the performance in various cases. The research team used this to test the predefined pipelines.

In the three PCD test cases, the linear and elbow type pipe mapping performance were compared with the manual work. The performances averaged 87.8% and 75.5%, respectively. As a result of the pipe object mapping pipeline performance test of large-capacity point cloud data, approximately 85% of pipe objects were extracted. The pipe object extraction performance improved by 104.4% between the proposed pipeline method and the software to be compared. Pipe extraction performance errors mainly occurred in small and complex pipes. The proposed method can precisely set the parameters of the Scan-to-BIM mapping pipeline, so it can obtain a precision that is difficult to set in the comparison software.

In case studies, the manual work is more accurate than the automatic method. However, the larger and more complex PCD is, the more human error occurs when adjusting the Scan-to-BIM work parameters every time. This can lead to rework costs and poor quality. The proposed method allows you to check the result of adjusting the operation parameters through the execution of the automated Scan-to-BIM pipeline quickly. Each pipeline component can be combined for its intended use, so it is easy to reuse the data processed in a specific step. For example, the pipeline can be reconfigured to extract geometry and BIM objects by reusing PCDs that have already been filtered in other software.

The Scan-to-BIM mapping pipeline structure can increase data processing performance by using parallel processing. For future research, it is planned to further improve the data processing speed by applying parallel processing. Various MEP projects are planned to be tested by improving Scan-to-BIM pipeline and parameter set. Also, the proposed method sets the Scan-to-BIM parameter according to the purpose. If there are enough ground true data-related parameter sets, the parameter setting process can also be automated by using AI (Artificial Intelligence) such as Deep Learning. This is one of the future research projects.

**Author Contributions:** T.K.—Software and algorithm development, Theory development and writing, Fund support, Use case and data development; S.P.—Software and algorithm development; K.K.—Research trend review, Methodology development; D.K.—Research trend review, Idea development; J.K.—Use case survey and data development. All authors have read and agreed to the published version of the manuscript.

**Funding:** This research was supported by KICT (Korea Institute of Civil Engineering and Building Technology) grant number [KICT 2020-0559] and KAIA (Korea Agency for Infrastructure Technology Advancement (KAIA) grant number [20AUDP-B127891-04].

**Conflicts of Interest:** The authors declare no conflict of interest.

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
