# Peer review of "Rule-Based Scan-to-BIM Mapping Pipeline in the Plumbing System"

_applsci, doi:10.3390/app10217422_

Round 1

Reviewer 1 Report

Dear Authors

The work presents typical framework for data processing related to scan-to-BIM (Building Information Modeling) for plumbing system. BIM is relatively novel technology applicable for  facility "from earliest conception to demolition" and facility management.

General remark:

- the paper seems to be more as technical report then scientific work. There is no discussion in the end and conclusions are very general without future works but fulfills the scope of Applied Sciences.

Detailed remarks, and questions:

- in grid generation stage an important parameter is grid size. What grid size did the Authors usually use to ensure enough detailed geometrical representation without significant lost of data,

- for outdoor data acquisition the Authors applied dron and photogrammetry. There is no information about key points detection and description algorithm, used software, how many pictures was analyzed,

- what are the units in which the results are presented in Table 6.

The better place for the work presentation would be  ICBIMBD 2020: 14. International Conference on Building Information Modeling and Building Design, December 28-29, 2020 in Paris, France.

https://waset.org/building-information-modeling-and-building-design-conference-in-december-2020-in-paris

Best regards

Author Response

Response to Reviewer 1 Comments

Dear Authors

The work presents typical framework for data processing related to scan-to-BIM (Building Information Modeling) for plumbing system. BIM is relatively novel technology applicable for  facility "from earliest conception to demolition" and facility management.

General remark:

Point 1: the paper seems to be more as technical report then scientific work. There is no discussion in the end and conclusions are very general without future works but fulfills the scope of Applied Sciences.

Response 1: Thanks for the comments. we revised the conclusion including discussion like below.

In case studies, the manual work is more accurate than the automatic method. However, the larger and more complex PCD, the more human error occurs when adjusting the Scan-to-BIM work parameters every time. This can lead to rework costs and poor quality. The proposed method allows you to check the result of adjusting the operation parameters through the execution of the automated Scan-to-BIM pipeline quickly. Each pipeline component can be combined for its intended use, so it is easy to reuse the data processed in a specific step. For example, the pipeline can be reconfigured to extract geometry and BIM objects by reusing PCDs that have already been filtered in other software.

The Scan-to-BIM mapping pipeline structure can increase data processing performance by using parallel processing. For future research, it is planned to further improve the data processing speed by applying parallel processing. Various MEP projects are planned to be tested by improving Scan-to-BIM pipeline and parameter set. Also, the proposed method sets the Scan-to-BIM parameter according to the purpose. If there are enough ground true data-related parameter sets, the parameter setting process can also be automated by using AI (Artificial Intelligence) such as Deep Learning. This is one of the future research projects.

Detailed remarks, and questions:

Point 2: in grid generation stage an important parameter is grid size. What grid size did the Authors usually use to ensure enough detailed geometrical representation without significant lost of data,

Response 2: We explained the comments related to grid size.

The parameter value must be able to be determined differently according to the purpose of Scan-to-BIM. This is because the dimensions and properties of the modeled BIM object are different depending on the use case.

After calculating the total size of the point group, the grid sizes divided by 3 on the X-Y-Z axis were used to generate voxels. Voxels were used to improve PCD calculation performance. Downsampling was performed for LoD processing, and the parameter value was 2 cm. For reference, the pipe radius required was between 20 and 50 cm. The difference in diameter between these pipes was 6 cm. When mapping the extracted geometries to BIM objects, they were converted into BIM objects using Scan-to-BIM mapping ruleset. In addition, the parameters such as grid size or LoD were chosen considering these situations.

Point 3: for outdoor data acquisition the Authors applied dron and photogrammetry. There is no information about key points detection and description algorithm, used software, how many pictures was analyzed,

Response 3: We explained the information including photogrammetry software like below.

For photogrammetry, 74 photos were taken using a DJI drone, and point clouds were created using Pix4D. The image resolution is 4000 x 3000 pixels.

Point 4: what are the units in which the results are presented in Table 6.

Response 4: We added the unit like below.

Table 6. Scan-to-BIM Mapping Pipeline Performance (count, %)

Point 5: The better place for the work presentation would be ICBIMBD 2020: 14. International Conference on Building Information Modeling and Building Design, December 28-29, 2020 in Paris, France.

https://waset.org/building-information-modeling-and-building-design-conference-in-december-2020-in-paris

Response 5: Thank you for good information. We will share related information with the research team.

In addition, some English sentences have been improved (marked in red).

Thanks for your comments.

Sincerely.

Reviewer 2 Report

This article presented a scheme for scan-to-BIM process, along with a case of practical implementation. The structure of the writing is good. The foundation of the proposed scheme is sound. Several suggestions are listed as following for the reference of authors.

  1. This scheme is designed to be operated as a semi-automated process. As outlined in the abstract, "If the Scan-to-BIM process can be formalized based on predefined rules, the repetitive process in various cases can be automated by re-adjusting only the parameters". While the rules and parameters are described in Chapter 4, could human intervention points also shown in the diagrams?
  2. After reading Chapter 4, this reviewer still found difficult to understand the rules. Are these rules generated automatically without human intervention? Is there a weighting/evaluation scheme for the rules?
  3. In line 75, Rhino is mentioned but not cited. Which version is the Rhino? Wouldn't it be fair to credit Rhino team with a proper citation?
  4. Lastly, this reviewer is interested in knowing the environment of the implementation. The description of "quad core and 4G memory" is too brief. Were GPU processing applied? Parallel processing is the request 4.

Author Response

Response to Reviewer 2 Comments

This article presented a scheme for scan-to-BIM process, along with a case of practical implementation. The structure of the writing is good. The foundation of the proposed scheme is sound. Several suggestions are listed as following for the reference of authors.

Point 1: This scheme is designed to be operated as a semi-automated process. As outlined in the abstract, "If the Scan-to-BIM process can be formalized based on predefined rules, the repetitive process in various cases can be automated by re-adjusting only the parameters". While the rules and parameters are described in Chapter 4, could human intervention points also shown in the diagrams?

Response 1: Thanks for your comments. The diagram including human intervention was described like below.

As shown in Figure 5, the modeler sets parameters to suit the purpose or reuses them in the parameters library. This is because the dimensions and properties of the modeled BIM object are different depending on the use case.

Figure 5. Purpose-oriented Scan-to-BIM Parameters Setting related to Pipeline Component

Point 2: After reading Chapter 4, this reviewer still found difficult to understand the rules. Are these rules generated automatically without human intervention? Is there a weighting/evaluation scheme for the rules?

Response 2:  Thanks for your comments. For Scan-to-BIM purposes, parameters (including weights) must be set by the modeler. It is not easy to automate this for all use cases (and thus the characteristics of input and output data such as PCD and BIM objects will all be different). Even in most commercial software, these parameter adjustments must be entered manually in each of the UI menus. In this regard, it has been described as follows.

The parameter value must be able to be determined differently according to the purpose of Scan-to-BIM. In other words, Scan-to-BIM configuration with parameter setting is purpose-oriented.

Also, the proposed method sets the Scan-to-BIM parameter according to the purpose. If there are enough ground true data-related parameter sets, the parameter setting process can also be automated by using AI (Artificial Intelligence) such as Deep Learning. This is one of the future research projects.

Point 3: In line 75, Rhino is mentioned but not cited. Which version is the Rhino? Wouldn't it be fair to credit Rhino team with a proper citation?

Response 3: The sentence was revised like below.

A semantic labeling study for Scan-to-BIM focused on creating a mesh using Rhinoceros and Grasshopper and then labeling BIM planar members using the geometric features of the mesh surface [8].

  • BASSIER, Maarten; VERGAUWEN, Maarten; VAN GENECHTEN, Bjorn. Automated Semantic Labelling of 3D Vector Models for Scan-to-BIM. In: Proceedings of the 4th Annual International Conference on Architecture and Civil Engineering (ACE2016). België, 2016. p. 93-100.

Point 4: Lastly, this reviewer is interested in knowing the environment of the implementation. The description of "quad core and 4G memory" is too brief. Were GPU processing applied? Parallel processing is the request 4.

Response 4: Regarding, it has been described as follows.

For reference, the computer specifications tested are Intel Core i5, quad-core and 4G memory without GPU.

In addition, some English sentences have been improved (marked in red).

Thanks for your comments.

Sincerely.

Round 2

Reviewer 1 Report

The reviewer would like to thank the Authors for their comprehensive answers.

Best regards